# Sensor-Integrated Chairs for Lower Body Strength and Endurance Assessment

**DOI:** 10.3390/s24030788

**Published:** 2024-01-25

**Authors:** Alexander W. Lee, Melissa S. Lee, Daniel P. Yeh, Hsi-Jen J. Yeh

**Affiliations:** 1Chino Premier Surgery Center, Chino, CA 91710, USA; alexanderlee8690@gmail.com; 2Walnut Valley Research Institute, Walnut, CA 91789, USA; melissalee2390@gmail.com (M.S.L.);; 3Department of Engineering and Computer Science, Azusa Pacific University, Azusa, CA 91702, USA

**Keywords:** aging, leg strength, chair stand test, sit-to-stand, strain gauge, weight sensor, multi-channel data, temperature compensation, weight distribution, internet of things

## Abstract

This paper describes an automated method and device to conduct the Chair Stand Tests of the Fullerton Functional Test Battery. The Fullerton Functional Test is a suite of physical tests designed to assess the physical fitness of older adults. The Chair Stand Tests, which include the Five Times Sit-to-Stand Test (5xSST) and the 30 Second Sit-to-Stand Test (30CST), are the standard for measuring lower-body strength in older adults. However, these tests are performed manually, which can be labor-intensive and prone to error. We developed a sensor-integrated chair that automatically captures the dynamic weight and distribution on the chair. The collected time series weight–sensor data is automatically uploaded for immediate determination of the sit-to-stand timing and counts, as well as providing a record for future comparison of lower-body strength progression. The automatic test administration can provide significant labor savings for medical personnel and deliver much more accurate data. Data from 10 patients showed good agreement between the manually collected and sensor-collected 30CST data (M = 0.5, SD = 1.58, 95% CI = 1.13). Additional data processing will be able to yield measurements of fatigue and balance and evaluate the mechanisms of failed standing attempts.

## 1. Introduction

Lower-body strength is an important factor for the overall health and quality of life of elderly adults [1]. Falls can cause catastrophic injuries in older adults, which can be very difficult to recover from [2]. Lower-body strength measurements can be used to assess the fall risks associated with aging. The Fullerton Functional Test developed by Jones and Rikli [3,4,5] contains two lower-body tests that use the ability of an individual to stand from a sitting position on a chair without the assistance of the hands to provide a quantitative measure of lower-body strength. The two Chair Stand Tests measure the time required to execute a fixed repetition of sit-to-stand and the number of sit-and-stands in a specified time. Specifically, the Five Times Sit-to-Stand Test (5xSST) measures the amount of time in seconds required for the subject to stand from a sitting position five times, while the 30 Second Sit-to-Stand Test (30CST) counts the number of sit-to-stand repetitions that the subject can execute in a 30-s period [6].

These tests have been correlated with fall risk [7], and standard performance metrics have been developed for various age groups. For example, for the 5xSST, there is the risk of recurrent falls for times greater than 15 s [8], and further assessment of fall risk is required for times greater than 12 s. Norms or averages for various age groups are also given in the references [8,9].

The chair stand tests are performed by the patient and guided by a clinician manually with a stopwatch. The clinician records the number of sit-to-stand cycles or the time to perform the test. This research sought to create an automatic method to accurately acquire the chair stand test data, reducing the workload of the clinician administering the test.

Several other methods have been proposed for the automated collection of chair stand data. Millor et al. [10] proposed using a specialized accelerometer and gyroscope attached to the subjects’ L3 lumbar spine to collect both movement and leaning data during the sit and stand transition. The method used wired technology and is thus not very portable. Cobo et al. [11] proposed an accelerometer attached to the thigh of the patient to measure movement and transmit the data via Bluetooth to a tablet. This approach uses wireless technology to maintain portability and takes into account the actual usage environment. Hellmers et al. [12] proposed a similar system using an IMU secured with a specialized inertial measurement unit (IMU, accelerometer plus gyroscope) belt and a separate ground reaction force plate to collect data when the user is standing. These types of systems require patients to wear sensors on their bodies, and clinicians must take additional steps to attach the sensors securely. This can add obtrusiveness to the patient and complications in real-world clinical settings.

Goncalves et al. [13] proposed using an infrared distance sensor mounted above the chair to measure the distance as the patient stands and sits and a microswitch to measure pressure on the seat. Takeshima et al. [14] proposed the use of infrared depth cameras to capture the motion of patients as they execute sit-to-stand movements. While not obtrusive to the patient, the overhead distance sensor mounted on a tripod column or infrared depth camera mounted on a tripod 3 m away can cause difficulty in setup and portability. In another paper, Cobo et al. [15] designed an ultrasonic sensor mounted on the back of the seat to measure the distance between the subject and the seat back. The authors reported noisy sensor signals that may degrade performance for older adults. 

In our research, we designed and built sensor-integrated chairs that use multiple sensors mounted on a commonly available chair. To maximize portability, we planned for wireless networking technology and battery operation to avoid tethering the sensor-integrated chairs, and we transmitted the collected sensor data to a cloud-based server. We designed the entire system as a single unit, so no sensors are attached to the patients and no sensors need to be set up external to the chair. 

## 2. Materials and Methods

The main goals of the design of the sensor-integrated chair are (1) the availability of components, (2) reduction in costs, and (3) the ease of exporting the design to other chairs. All of these factors facilitate the integration of the design into existing chairs that may function well in various facilities and be preferred by different clinicians. Therefore, the design focuses on using commonly available components and avoiding specialized or custom parts that add costs and are difficult to obtain.

The bill of materials is presented in Table 1.

### 2.1. Sensor Integration 

To meet the above goals, the sensor-integrated chair uses strain-gauge weight sensors mounted to the bottom of the legs of a commonly available chair (Figure 1). This design is an improvement on previous designs [16], which placed weight sensors under the seating area. Placing the sensors under the seating area requires more extensive modification in that it involves the removal of the seating area and mounting the weight sensors. The seating area needs to be supported entirely by the weight sensors, which causes it to be free of mechanical constraints, and the entire chair lacks stability.

Placing the sensors on the bottom of the legs has its challenges, as there needs to be a mechanical assembly to hold the stationary part of the weight sensors in place while allowing the movable part to have free motion to provide accurate deflection and strain measurement. For the current implementation, we chose a readily available mechanical structure, using the feet of commercial human body weight scales or bathroom scales. The feet are optimized for human body weight ranges, and each scale typically comes with 4 feet. Mounting the scale feet on the chair legs requires matching the chair legs’ diameters to the scale feet. Therefore, the chair was first fixed onto a rigid board, and then, the feet were attached to the opposite side of the rigid board directly under each of the legs. 

The chair must be mounted on the board so the board does not protrude excessively in front of the chair. This design ensures that the subject does not place their feet on the board while sitting and that the subject’s legs remain clear of the board while performing the sit-to-stand movements. Doing so would compromise the weight measurement as the user’s weight would be borne (partially) by the board. We determined that a 1 inch (2.5 cm) margin is acceptable. In addition, mounting the board and the weight scale feet to the legs of the chair adds height to the chair. The legs of the chair may need to be shortened to maintain a 17 inch (43 cm) seat height as specified in the Chair Stand Tests.

Within each of the four scale feet is a weight sensor designed so that the part of the foot in contact with the ground applies the weight force to the movable portion of the sensor. Each weight sensor consists of two resistive elements (strain gauge) deposited on opposite sides of the movable portion, nominally at 1 kΩ. When the movable portion is deformed or strained from the applied force, one strain gauge is compressed, decreasing its resistance, while the other strain gauge is stretched, increasing its resistance. For small strains, the strain is linearly proportional to stress, and the resistance change is linearly proportional to the strain. 

For weight scale applications, only the total weight is required. Here, the four sensors in the four feet are wired together in a single Wheatstone bridge configuration that sums the changes in resistance when the voltage difference is measured across the bridge. In this case, one amplifier and one analog-to-digital converter (ADC) are required to obtain the sum of the weights on the four feet. 

In our application, we want to obtain four independent weight readings from each of the four sensors. For four independent weight channels, each weight sensor must connect independently to its own signal path (amplifier plus ADC). In this configuration, two 1 kΩ resisters are typically used to form the other arm of the Wheatstone bridge. 

During testing, we found that the weight sensor plus two resistor configuration is sensitive to temperature changes, causing a significant drift in the weight reading over time as the temperature changes. We did not discover other environmental sensitivities, such as sensitivity to light. Although relative humidity can cause drift to strain gauge sensors, humidity changes are typically slow relative to the period of data capture; we compensate for any long-term drift by calibrating the sensors before each measurement.

We experimented with different designs to compensate for the temperature drift. We found that using an unweighted 1 kΩ weight sensor [17] as the other arm of the Wheatstone bridge (Figure 2), instead of using two 1 kΩ resistors, effectively compensated for the temperature drift.

We chose an ASIC (application-specific integrated circuit) optimized for human weight range measurements for the amplifier and ADC. Weight-measuring ADCs have different requirements compared to audio or other types of ADCs. Weight sensors are rated by their full measurement range, the maximum force that can be applied before significant nonlinear response or damage to the sensor. Using stiffer (higher modulus) material for the movable part results in smaller strain from a given stress and, therefore, a larger measurement range. Human body weight sensors are typically designed for a 50 kg range, for a total weight measure of 4 × 50 kg = 200 kg with 0.1 kg resolution. Since the change in strain is very small at the full rated measurement range, the ADC must have very good resolution and low noise. However, since they measure static weight, the measurement rate of weight scales can be low, on the order of ~1 Hz. With 10 sample averaging, the sample rate of the ADC is in the order of ~10 sps (samples per second).

We chose the commercially available HX711 ASIC from Avia Semiconductors [19] for the initial system due to its high commercial availability. The HX711 is an integrated amplifier and ADC chip with a selectable sample rate of 10 sps or 80 sps and a resolution of 24 bits with an effective number of bits (ENOBs) of 20 or lower. For ease of connection, we used HX711 ICs packaged in module form. The module consists of an analog end that powers the Wheatstone bridge and reads the differential signal and a digital end that provides a 2-wire protocol to the host microcontroller or CPU.

Other signal chain components can be used as well, including the NAU7802 ASIC from Nuvoton [20] and the ADS1231 ASIC from Texas Instruments [21], both of which operate at similar specifications (24-bit ADC at 10 sps or 80 sps). The TI ADS1222 [22] provides a 24-bit ADC with 20-bit ENOBs at 240 sps, which can be used if a higher sample rate is desired. However, we found 80 sps to be sufficient for the sit-to-stand application.

The digital end of each HX711 module is connected to two pins on the microcontroller. The digital interface is not I2C compliant, and a custom driver is used. However, due to the use of the custom driver whose pseudocode and timing are detailed in the datasheet, it became simpler for us to create an asynchronous multi-channel (multi-chip) driver used in this 4-chip application.

### 2.2. Microcontroller Platform 

A data interface needs to be provided to transfer data collected from the sensors integrated into the chair in real time. For convenience of operation, we chose a wireless interface, specifically WiFi. An additional advantage of a microcontroller that supports WiFi is that the device can upload data directly to the cloud using a ubiquitous and existing WiFi infrastructure without the need for any bridge devices, as would be required if, for example, Bluetooth or Zigbee were used. With appropriate software development, the microcontroller can collect data, connect to the WiFi router, and upload the data using a variety of communication protocols without any additional hardware.

We chose the WiFi-integrated microcontroller, the ESP32 system-on-chip (SOC) from Espressif Systems [23]. Although it has relatively low memory size and computing power compared to modern CPUs, it offers excellent performance for an embedded microcontroller. It is one of the workhorses for Internet of Things (IoT) applications [24]. The ESP32 in module form provides dual 240 MHz 600 DMIPS CPUs, one for WiFi and the other for user firmware. It implements the full TCP/IP stack, allowing it to work as a fully stand-alone WiFi node. We used the Arduino integrated development environment (IDE) due to its ease of use and extensive library of useful open-source libraries. The programming language is C++.

Because each HX711 ADC module has its own oscillator, the data acquisition rate among the four sensors is not synchronized. We configure the HX711 modules to run 80 sps. In initial testing, the actual sample period ranged between 11 ms and 12 ms, exceeding the 80 sps specification. To handle the distribution in sample rates, we developed a custom driver firmware to asynchronously gather data from each sensor as soon as they were ready. The firmware then reports the data from all four sensors synchronously at a fixed sampling period not exceeding 12 ms.

We calculated and stored the scaling constant between the ADC reading and weight to calibrate the weight sensor ADC signal chain. The scaling constant is obtained by dividing the difference between the ADC readings with (gross reading) and without (tare reading) a known weight placed on the chair by the known weight. After one-time calibration, the object’s (net) weight equals the ADC reading with that object on the chair minus the ADC reading with no object divided by the scaling constant. To confirm that the scaling constant is consistent for each of the four independent weight sensors—the ADC signal chains—we placed a known weight at different locations on the seat of the chair for different weight distributions on the legs. Although the four sensors registered different readings as the weight was shifted, the combined total weight remained the same within the resolution limits, indicating consistent scaling constants.

### 2.3. Software and System

Both a legacy menu-driven serial interface and a browser interface (Figure 3) were developed as the ESP32 firmware to facilitate the data collection process. The browser interface utilizes MQTT Subscription [25] to receive and execute commands sent from a browser. Parameters can be set, including the sampling period (and therefore the sample rate) and the trial period (number of seconds to collect the data). We recommend the trial period to be ~40 s for 30CTS to allow ample time before and after the 30-s test, and ~20 s for 5xSST since values above 14 sec are considered a positive indication of high fall risk. Additional parameters such as subject number (or user ID) and trial classification (such as subject physical state to be specified by the clinician) can be specified.

Before each trial run, consisting of one repetition of the 30CTS or 5xSST, the system will determine the offset when no weight is placed on the chair and tare the system. After tare offset, which takes less than one second, the user interface will instruct the clinician to commence the trial. 

Data from the four sensor channels are continuously collected during the trial at the specified sampling period. At the end of the trial, additional data processing is performed to provide the time series total weight on the chair and calculate the sit-to-stand and stand-to-sit transition times. Various peak detection and digital filtering algorithms were attempted for transition detection. However, graphing typical sit-to-stand transition data indicates a square wave or a truncated sine wave pattern; after a successful sit-to-stand transition, the measured total weight moves quickly to and remains cleanly at the tare value (zero reading) as the subject removes all weight from the chair. After a stand-to-sit transition, the signal is not as clean and exhibits mechanical oscillation or bounce. Thus, we developed a fast sit-to-stand transition algorithm using the mean of the total weight data series as the crossing threshold. This fast and simple algorithm has been shown to be very accurate. The number of sit-to-stand transitions can be reported immediately to the clinician for redundant recording. Since the data are collected and stored, more sophisticated data analysis can be performed offline.

The data from the sensor channels, in addition to the parameters or metadata, are uploaded to a cloud-based server. We chose the JSON (JavaScript Object Notation) [26] data format, using either MQTT Publish or HTTP POST protocols (Table 2). Due to the limited memory of the ESP32 module and to ensure reliability, the metadata and data from each channel are packetized and transmitted in separate publish or POST requests.

Regarding data storage and retrieval, the system stores data from each trial on a cloud-based server, and the stored data can be retrieved with HTML pages. The HTTP interface also provides graphs of the separate weight channels, total weight, sit-to-stand transitions, and rate of weight change. 

The data collection server consists of the following modules: (1) HTML pages that serve as the browser interface from the clinician to the ESP32 firmware, (2) the MQTT Subscription and HTTP POST endpoint to receive the data from the ESP32 and store them on the server, and (3) HTML pages to perform data access, processing, and graphing. The browser interface converts clinician commands to MQTT Publish requests matching the MQTT Subscription topics on the ESP32 firmware and displays messages from the ESP32 firmware by MQTT Subscriptions. The collected data are stored in text files whose names consist of the user ID and the sequence number of that user ID in JSON format. The initial data collection server software was developed in Python and ported to Node.js and PHP. The various software ports give great flexibility in the different cloud hosting platforms where the server software can be deployed.

The stored data can be retried or processed via additional HTML pages. We currently implement the sit-to-stand transition times using the same thresholding algorithm for immediate feedback to the clinician and the rate of weight change (or weight velocity) on the seat by calculating the numerical derivative of the time series weight.

## 3. Results

### 3.1. Functional Testing

We successfully obtained four independent data channels in a 12 ms sampling period (~83 sps) for the right-back, right-front, left-front, and left-back legs of the chair. The data were collected from a 53-year-old male with no lower-body physical impairment. We summed and obtained the total weight from the four independent channels and determined the sit-to-stand event times. We also calculated the rate of weight change, or weight velocity, which can indicate the strength with which the subject leaves the chair.

In measuring independent strain-gauge weight sensors, we successfully mitigated the weight sensors’ temperature drift by using unloaded temperature-compensating weight sensors. We saw no noticeable temperature drift during the collection period. Also, we determined that the calibration constants of the four independent signal chains were consistent within the resolution limits. The firmware we developed was capable of collecting data from each sensor at full speed asynchronously and providing synchronous readings at the desired sample rate. 

From the graph of various sit-to-stand transitions (Figure 4), we measured mechanical oscillations between 4 Hz and 5 Hz. We also found that sit-to-stand transition occurred over seven sample points or more (from 90% to 10% sitting weight). Therefore, an ~80 sps ADC sample rate is adequate for this application. However, the recommended minimum sampling rate for other body motion systems, such as ground reaction force plates, is 200 sps to capture jumping motion [27]. We will be able to use faster ADCs at a higher cost if the need arises during analysis of the clinical data.

### 3.2. Validation 

We completed the clinical data collection process with the sensor-integrated chair at the Chino Hills Premiere Surgery Center, a pain management clinic. We present the validation results of the chair stand tests for 20 patients. The age of the patients ranged from 26 to 84 years old (M = 61.5, SD = 13.7), and the weight of the patients ranged from 97.0 to 299.4 lb (M = 193.3, SD = 45.3). The manually measured 30CST ranged from 3 to 18 (M = 8.78, SD = 4.0), and the 5xSST ranged from 5 to 42 s (M = 20.6, SD = 9.9). In contrast, the sensor recorded a 30CST range from 4 to 15 (M = 7.9, SD = 3.2). The sensor measuring the 5xSST ranged from 12 s to the maximum of 30 s, at which time data collection was terminated because anything above 14 s was already considered a failure; in total, 4 out of the 20 patients’ data reached the cutoff (M = 19.9 and SD = 5.8). 

To assess the performance of the sensor-integrated chair, we calculated the difference between the manual and sensor measurements. The performance of the sensor-integrated chair for the 30CST was good (M = 0.9, SD = 1.29, 95% CI = 0.65). For the 5xSST, after removing data that exceeded the 30-s cutoff, we obtained reasonable performance (M = −2.66 s, SD = 2.27, 95% CI = 1.21).

## 4. Discussion

The sensor-integrated chair passed functional testing and captured four independent sensor data channels, one under each leg of the chair. We fixed a board on the bottom of the chair to provide structural rigidity, which did not interfere with data collection. We plan for further design optimization by eliminating the board. The data are captured at a sufficient rate for sit-and-stand application at 80 sps, although we can move to a higher sampling rate if the conditions require. The temperature compensation method to mitigate the effect of temperature drift using an additional sensor per channel was successful.

The captured data are transmitted either as an HTTP POST or an MQTT Publish request to the server, in JSON data format. The web-based server contains an HTML interface to control the data collection process and retrieve or graph the collected data with a user-friendly interface. Functional testing was completed, and the captured data were verified to be accurate.

We analyzed preliminary clinical data from 20 patients at a pain-management clinic. We found a good correlation between the manually collected and sensor-collected data for the 30SST and a reasonable correlation for the 5xSST. We are continuing data collection, and we will further analyze the clinically collected sit-to-stand timing and counts with human-collected data and make improvements to the data processing algorithm.

With the time series weight data, we plan for future work with more sophisticated data analysis than just the raw times or raw counts. For example, change in or lengthening of the period between successive sit-to-stand transitions can indicate the level of fatigue and estimate endurance as the subject performs the tests. The rate of change in weight or weight velocity can indicate the force of the stands and assess the strength of the musculature (Figure 5). 

Although we use the time series total weight as the subject performs the chair stand tests, we designed the system to capture four independent channels. For example, the relative balance between the right and left leg can provide insight into left versus right favoring in injury recovery. The shift in weight from the front to the back immediately before standing can give insight into lower limb weakness and the subject rocking to build momentum to assist in standing. 

## 5. Conclusions

This paper presents a sensor-integrated chair for the automatic data capture of the 5xSST and the 30CST which aims to reduce the administering clinicians’ workload. This system is designed so that sensors are integrated into the chair, rather than worn by the patient, to minimize obtrusiveness in clinical data collection. Because no sensing element needs to be set up external to the chair, setup is greatly simplified. Additionally, the system transmits data via wireless communications and is powered by a battery pack to avoid tethering and maximize portability. A set of web pages was developed to provide a convenient user interface to run tests and capture data.

The sensor-integrated chair utilized commercially available components to ensure wide availability, low cost, and ease of porting into other chairs. We believe these choices facilitate the integration of the design into existing chairs that function well in various facilities and are preferred by different clinicians. After successful functional testing and clinical validation, we believe that this research could be beneficial in assessing lower-body strength and fall risk in older adults.

## Figures and Tables

**Figure 1 sensors-24-00788-f001:**
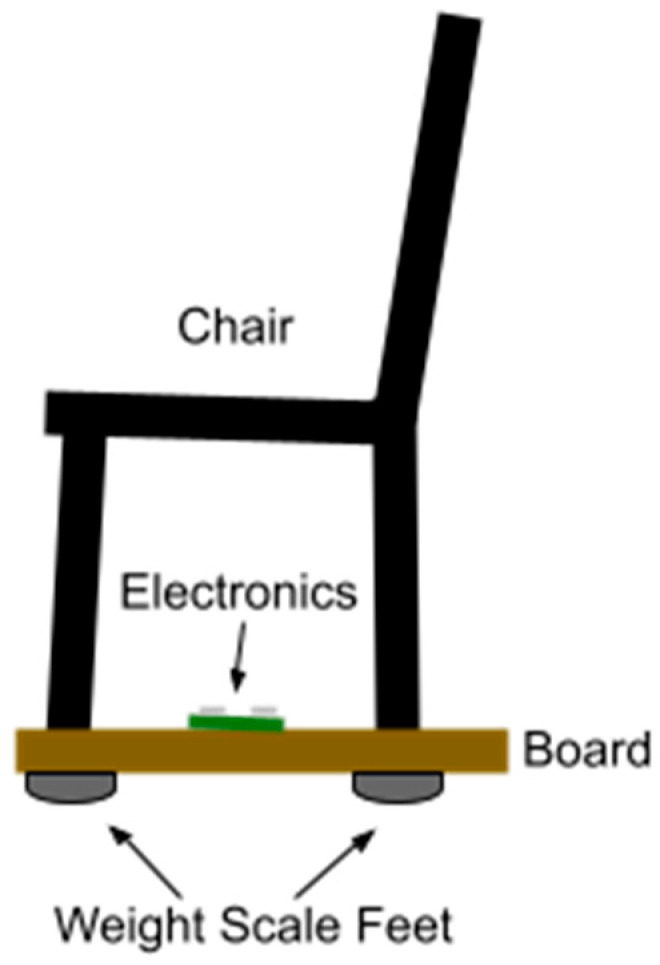
Drawing of the sensor-integrated chair. Four (4) weight scale feet are mounted directly under the chair legs on the other side of the board, each containing a 50 kg weight sensor. The electronics consist of 4× analog-to-digital converters, 4× temperature-compensating weight sensors, the WiFi-enabled microcontroller, and a USB battery pack for power. Each weight scale foot is connected to its respective ADC in a Wheatstone bridge configuration with a temperature-compensating weight sensor.

**Figure 2 sensors-24-00788-f002:**
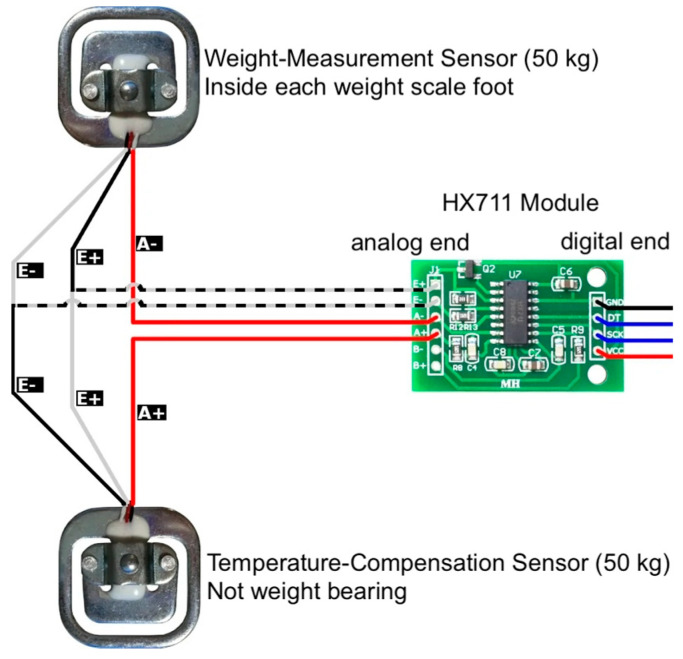
Connection diagram of a weight sensing channel with temperature compensation. The analog end of the HX711 module contains the excitation or input terminals (E+ and E−) and the signal or output terminals (A+ and A−) of the Wheatstone bridge. One arm of the bridge consists of the upper and lower strain gauge resistors of the weight-measuring sensor, and the other arm consists of the lower and upper strain gauge resistors of the temperature-compensating sensor. The reversal of the connections between the sensors provides compensation for the temperature drift. (Diagram modified from the ESP Easy Document site [18]).

**Figure 3 sensors-24-00788-f003:**
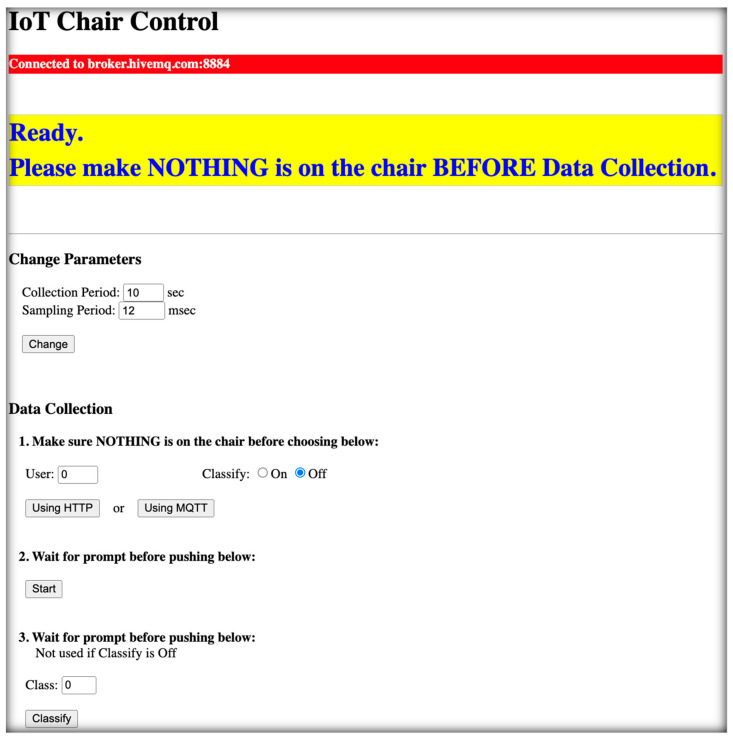
Data collection HTML interface. The top bar (in red) shows the connection status to the MQTT broker. The second bar (yellow) displays the MQTT messages from the microcontroller of the sensor-connected chair. The rest of the page provides the user interface for changing the data collection parameters and collecting data. Data collection is performed in a three-step process: (1) specifying the user (subject), whether to classify the trial, and the data upload method; (2) waiting for offset calibration (tare) to complete with no weight on the chair, and then start the trial; and (3) classification of the trial, to be used for machine learning at a post-data collection stage, if classify is turned on.

**Figure 4 sensors-24-00788-f004:**
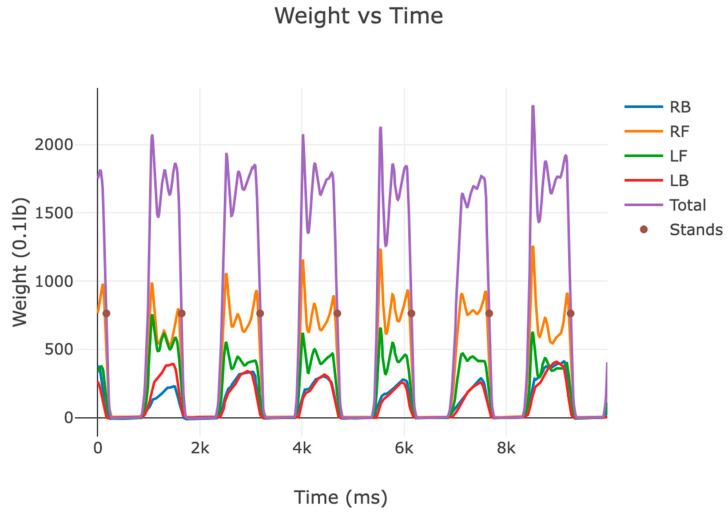
Graph of collected data. This graph shows the time series weights of the four independent (right-back, right-front, left-front, and left-back) channels, the total weight, and the calculated times of the sit-to-stand transitions (stands). The graph shows seven (7) stands in a 10 sec interval. There are 4.5 Hz mechanical oscillations when the subject sits on the chair and no oscillations when the subject stands.

**Figure 5 sensors-24-00788-f005:**
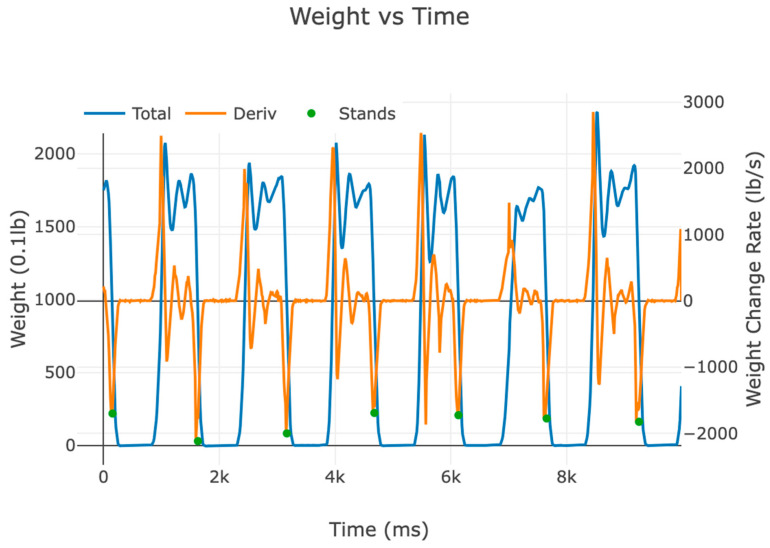
Graph of weight velocity. This graph shows the total weight, the weight rate of change (deriv.), and the sit-to-stand transitions (stands) times. The magnitude of the weight velocity indicated by the y-axis location of the dots (1700~2100 lb/s) can provide an estimate of the strength of the stand.

**Table 1 sensors-24-00788-t001:** Bill of materials of the sensor-integrated chair.

No.	Qty.	Part	Description
1	1	Chair	Straight back, without armrest, seat 43 cm (17 in)
2	1	Board	50 cm × 50 cm × 2 cm (20 in × 20 in × ¾ in)
3	1	ESP32-WROOM-32	Module with USB Port
4	1	USB Battery Pack	Supply 5V power to ESP32
5	4	Weight Scale Feet	Removed from bathroom scale, contains weight sensor
6	4	HX711	Module with analog end and digital end
7	4	50-kg weight sensor	For temperature compensation

**Table 2 sensors-24-00788-t002:** Data format in JSON. The data are sent in separate MQTT Publish or HTTP POST requests. The initial request contains parameters (metadata) regarding the trial, including the user (subject) ID *uid*, classification *cls* if any, trial period *Ttrl* in ms, sample period *Tsmp* in ms, and the number of channels, which is 4. This request is followed by data from the four channels, with the data channel label (for example, right-back or RB) and the time series *data_c_i* (channel *c*, sample *i*). Each channel has data length *n* = *Ttrl*/*Tsmp*. The separate uploads are merged into a single JSON statement.

**Data Parameters**
{“metadata”: {“user”: *uid*, “class”: *cls*, “T_trial”: *Ttrl*, “T_sample”: *Tsmp*, “N_dev”: 4}}
**Data Set**
{“data”: {“label”: “RB”, “data”: [*data_0_1, …, data_0_n*]}}
{“data”: {“label”: “RF”, “data”: [*data_1_1, …, data_1_n*]}}
{“data”: {“label”: “LF”, “data”: [*data_2_1, …, data_2_n*]}}
{“data”: {“label”: “LB”, “data”: [*data_3_1, …, data_3_n*]}}

## Data Availability

The data presented in this study are available from the corresponding author upon request.

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
