# Peer review of "Sensor-Integrated Chairs for Lower Body Strength and Endurance Assessment"

_sensors, 2024, doi:10.3390/s24030788_

Round 1
Reviewer 1 Report
Comments and Suggestions for Authors
The research has proposed an automated and efficient design for measuring Fullerton Functional test, using temporal weight-sensor data. The research question is not clearly stated neither in abstract nor introduction. The introduction section should be re-written with clear contribution statement.
The research problem this study has addressed is to design and develop an automated way of assessing lower-body strength of the older-adults.
The idea presented is interesting in a sense that this work uses a chair instead of other instrumental arrangement. The work addresses the gap with a very specific arrangement. Yet, there are work available on use of sensors (unobtrusive) for estimating fall risk, lower-body strength. For example, this below work is related but not discussed in this paper.
Takeshima, N., Kohama, T., Kusunoki, M., Fujita, E., Okada, S., Kato, Y., Kofuku, K., Islam, M.M. and Brechue, W.F., 2019. Development of Simple, Objective Chair-Standing Assessment of Physical Function in Older Individuals Using a Kinect™ Sensor. The Journal of Frailty & Aging, 8, pp.186-191.
The main proposal of this paper is laid over electrical circuit design which is more relevant for the engineering community. The work has chosen sensor-based chair as the case and presented a design and associate findings on the design testing, which would be of interest to the relevant community.
More evidences required to demonstrate the importance and use of these types of sensor-based chairs in hospitals and clinics. This paper added references to unpublished work [Ref #11], which should be removed. The authors should refer to a similar type of work that has been published. The work presented in this paper is more like designing artefact, which has lesser research contribution.
The experimentation/testing is currently done based on different sensor configurations. While the paper is trying to measure weight for different set-ups, the author should also consider about other factors, e.g., movement, time between stand and seat etc. Many design (or setup) decisions are presented in ad-hoc manner, without adding supportive evidence for them.
The paper founds the influence of temperature on the weight measurement, which has been addressed by one particular setup of the sensors. The author should consider other environmental biases (e.g., air, light) for the sensory design. Currently the paper does not provide any information on the subjects on which the chair and setup are tested. It would be good to test subjective biases – e.g., ages, health conditions, gender, to make the product/artifact more generic and robust.
The results section currently is very thin. The authors can consider to more some more analysis and presents the outcomes. Can the prototype be tested against other similar types of physical activities?
This paper does not have defined and clear research questions. The result section indicates the success of the design of the sensory system. The results section should present some comparison of the performance of similar system commercially available (or previously presented with research). The discussion section discussed future planning.
This paper currently does not have a literature review section, which should be added. Any reference which is unpublished work should be removed. Also, need to edit the section titles and numbering.
Reviewer 2 Report
Comments and Suggestions for Authors
The work presented an automated method to conduct Chair Stand Test of the Fullerton Functional Test. The authors developed a sensor-integrated chair to automatically capture dynamic weight of participants and collect the weight-sensor data. This paper should be improved according to the suggestions before publication.
1) The paper would be strengthened by providing more specifics on how thorough validation of the presented method was carried out, going beyond basic empirical testing.
2) The authors need to add more background information and related works. They should review and cite more studies related to their work. This will help set the context and show why their research is important.
3) The authors need to review and restructure the organization of the current manuscript as it appears section 2 encompasses more than one section.
4) While this work has presented some valuable ideas and findings, the authors need to provide a more self-critical perspective by explicitly discussing its limitations.
5) The paper is currently missing a conclusion section. The authors need to add an appropriate conclusion that effectively sums up the key points and takeaways from this research.
6)The authors should discuss some ideas for future work and next steps.
Round 2
Reviewer 1 Report
Comments and Suggestions for Authors
Thank you for addressing the comments. Understandably, some of the research recommendations provided earlier need a future data collection and analysis iteration. The current adjustments/updates have increased the merit of the work in an adequate manner.
Author Response
Thank you very much. We appreciate your comments very much!
Reviewer 2 Report
Comments and Suggestions for Authors
The authors have addressed all the previous comments.
Author Response

(The authors gave the same response as above.)
